# RadDiff: Retrieval-Augmented Denoising Diffusion for Protein Inverse Folding

## Abstract

Protein inverse folding, the design of an amino acid sequence based on a target 3D structure, is a fundamental problem of computational protein engineering. Existing methods either generate sequences without leveraging external knowledge or relying on protein language models (PLMs). The former omits the evolutionary information stored in protein databases, while the latter is parameter-inefficient and inflexible to adapt to ever-growing protein data. To overcome the above drawbacks, in this paper we propose a novel method, called retrieval-augmented denoising diffusion (RadDiff), for protein inverse folding. Given the target protein backbone, RadDiff uses a hierarchical search strategy to efficiently retrieve structurally similar proteins from large protein databases. The retrieved structures are then aligned residue-by-residue to the target to construct a position-specific amino acid profile, which serves as an evolutionary-informed prior that conditions the denoising process. A lightweight integration module is further designed to incorporate this prior effectively. Experimental results on the CATH, PDB, and TS50 datasets show that RadDiff consistently outperforms existing methods, improving sequence recovery rate by up to 19%. Experimental results also demonstrate that RadDiff generates highly foldable sequences and scales effectively with database size.

## 1 Introduction

Proteins are the molecular machines of life, executing a vast array of biological functions dictated by their three-dimensional (3D) structures (Koehler Leman et al., 2023). A grand challenge is the design of novel proteins with desired functions, a task for which protein inverse folding serves as a fundamental approach. The goal of protein inverse folding is to computationally design an amino acid sequence that will fold into a specified 3D backbone structure (Ingraham et al., 2019).

Recent advances in deep learning have shown great promise for protein inverse folding (Ingraham et al., 2019; Jing et al., 2020; Hsu et al., 2022; Fu & Sun, 2022; Dauparas et al., 2022; Gao et al., 2022b; Tan et al., 2023). Representative methods include denoising diffusion models (Yi et al., 2023; Bai et al., 2025), which have demonstrated a remarkable ability to generate high-fidelity sequences by learning the complex relationship between protein structure and sequence. However, these models often operate *de novo*, generating sequences conditioned only on structural geometry (Mahbub et al., 2025). This process omits the evolutionary information stored in large protein databases. Designing sequences without reference to known biological prior may lead to sequences that are biologically suboptimal (Huang et al., 2024), and it failed to leverage decades of collected protein data.

Recognizing the value of this external knowledge, some methods have successfully improved the protein design performance by incorporating information from large pre-trained protein language models (PLMs) (Zheng et al., 2023; Gao et al., 2023; Wang et al., 2024). While effective, these methods suffer from two key drawbacks. First, the PLMs often contain billions of parameters (Hayes et al., 2025), leading to a parameter-inefficient model for protein design. Second, this strategy produces a static knowledge base, which compresses the data into the fixed model parameters. Incorporating new protein data from continuously growing protein databases requires retraining the entire PLM, which is both inflexible and computationally prohibitive.

To address these challenges, in this paper we propose a novel method, called retrieval-augmented denoising diffusion (RadDiff), for protein inverse folding. The main contributions of our work are summarized as follows:

- We design a hierarchical search strategy that efficiently retrieves structurally similar proteins from large protein databases, providing flexible access to external knowledge.
- We introduce a residue-level alignment mechanism that constructs position-specific amino acid profiles, which serves as an evolutionary-informed prior and integrated into the denoising process via a lightweight module.
- Experimental results on the CATH, PDB, and TS50 datasets show that RadDiff consistently outperforms existing methods, improving sequence recovery rate by up to 19%. Experimental results also demonstrate that RadDiff generates highly foldable sequences and scales effectively with database size.

## 2 RELATED WORKS

### 2.1 PROTEIN INVERSE FOLDING

Protein inverse folding has been extensively explored for years. Traditional methods, such as Rosetta (Alford et al., 2017), frame the problem as a physics-based energy minimization task. More recently, deep learning based methods have shown great promise for this problem (Wang et al., 2018; Ingraham et al., 2019; Jing et al., 2020; Qi & Zhang, 2020; Fu & Sun, 2022; Dauparas et al., 2022; Gao et al., 2022b; Yi et al., 2023; Wang et al., 2024; Qiu et al., 2024). Graph Neural Network (GNN) based methods like GVP (Jing et al., 2020), ProteinMPNN (Dauparas et al., 2022), and PiFold (Gao et al., 2022b) demonstrate the power of learning representations directly from 3D protein structures to predict amino acid identities. Models such as LM-Design (Zheng et al., 2023) and KW-Design (Gao et al., 2023) leverage pre-trained PLMs to inject evolutionary information into the design process, significantly boosting performance. Diffusion-based methods like GradeIf (Yi et al., 2023) and MapDiff (Bai et al., 2025) have also shown great potential for modeling the conditional sequence distribution. PRISM (Mahbub et al., 2025) also employs the concept of retrieval-augmentation. However, the core methodologies of PRISM and our RadDiff differ significantly. PRISM operates at the embedding level, relying on pre-trained structure and sequence encoders to retrieve and integrate learned representations. In contrast, RadDiff's retrieval process is based on a direct comparison of 3D coordinates. Furthermore, RadDiff conditions the generation process on a position-wise amino acid profile derived from residue-level structural alignments, which provides a more direct form of guidance for the model.

### 2.2 PROTEIN STRUCTURE RETRIEVAL

Protein structure retrieval aims to retrieve similar protein structures from a large protein structure database given a query structure. We introduce two classical protein structure retrieval methods, TM-align (Zhang & Skolnick, 2005) and FoldSeek (van Kempen et al., 2022), which will be used in our method.

TM-align (Zhang & Skolnick, 2005) is a sequence-independent protein structure comparison tool. TM-align uses heuristic dynamic programming to find the optimal structure alignment between two structures based on template modeling score (TM-score) (Zhang & Skolnick, 2004). TM-score is a score function to measure the structure similarity between two protein structures, which has a value in (0,1], and 1 indicates that two structures are perfectly matched. TM-score>0.5 indicates that two structures are highly likely to share similar topology (Xu & Zhang, 2010). We will use the version US-align (Zhang et al., 2022) in our method, which is an extension of the TM-align which can generate more accurate structural alignment. Although TM-align and US-align are accurate to identify the similarity of two structures, it is time-consuming and infeasible to perform large-scale comparison for millions of or billions of structure pairs.

FoldSeek (van Kempen et al., 2022) is a fast protein structure retrieval method based on structural alphabet. FoldSeek discretizes structures into 3D interaction (3Di) sequences and uses MM-seqs2 (Steinegger & Söding, 2017) for ultra-fast retrieval, achieving search speeds several orders of magnitude faster than traditional alignment-based methods like TM-align. However, due to the

information loss in the discretization to a structural alphabet, FoldSeek is generally less accurate than TM-align (Litfin et al., 2025).

# 3 METHODS

## 3.1 PRELIMINARIES: DISCRETE DENOISING DIFFUSION

We follow the discrete denoising diffusion settings in Austin et al. (2021). In our setting, the vocabulary of protein inverse folding contains $K$ kinds of natural amino acids (i.e., $K = 20$). Given the initial amino acid sequence $\mathcal{S}$ (with $N$ amino acids), we denote its amino acid features as $\boldsymbol{X}^{aa} \in \mathbb{R}^{N \times K}$, where each row corresponds to an amino acid.

**(I) Forward diffusion process** The forward diffusion process, $q$, progressively corrupts an initial clean $\boldsymbol{X}_0^{\text{aa}}$ over $T$ timesteps. This creates a Markov chain of increasingly noisy sequences $\boldsymbol{X}_0^{aa}, \boldsymbol{X}_1^{aa} \ldots, \boldsymbol{X}_T^{aa}$. The transition at each step is defined by a matrix $\boldsymbol{Q}_t$, such that $q(\boldsymbol{X}_t^{aa} \mid \boldsymbol{X}_{t-1}^{aa}) = \boldsymbol{X}_{t-1}^{aa}\boldsymbol{Q}_t$. We use a standard cosine noise schedule $\beta_t$ to define a uniform transition matrix $\boldsymbol{Q}_t = (1 - \beta_t)\boldsymbol{I} + \beta_t \mathbf{1}_K \mathbf{1}_K^\top / K$, $\mathbf{1}_K$ denotes the all-one vector with dimension $K$. The final state $\boldsymbol{X}_T^{aa}$ converges to a uniform distribution over all amino acids, independent of the input $\boldsymbol{X}_0^{aa}$. For any noisy state $\boldsymbol{X}_t^{aa}$, it can be sampled in a closed form: $q(\boldsymbol{X}_t^{aa} \mid \boldsymbol{X}_0^{aa}) = \boldsymbol{X}_0^{aa}\bar{\boldsymbol{Q}}_t$, where $\bar{\boldsymbol{Q}}_t = \prod_{k=1}^{t} \boldsymbol{Q}_k$.

**(II) Denoising network and training objective** The goal is to learn the reverse process, $p_\theta(\boldsymbol{X}_{t-1}^{aa} \mid \boldsymbol{X}_t^{aa})$, to denoise a sequence from $\boldsymbol{X}_T^{aa}$ back to a clean sequence $\boldsymbol{X}_0^{aa}$. This is achieved by training a denoising network to predict the original clean sequence $\hat{\boldsymbol{X}}_0^{aa}$ given the noisy sequence $\boldsymbol{X}_t^{aa}$ at timestep $t$, along with our conditioning information. The network is trained to minimize the cross-entropy loss between its prediction and the true clean sequence.

**(III) Reverse denoising process** The reverse denoising process generates a new amino acid sequence by iteratively applying a denoising step for $t = T, \ldots, 1$. The generative distribution for this reverse transition, $p_\theta(\boldsymbol{X}_{t-1}^{aa} \mid \boldsymbol{X}_t^{aa})$, is estimated using the trained denoising network. Specifically, we marginalize over the network's predictions for the clean sequence, $\hat{p}_\theta(\boldsymbol{X}_{t-1}^{aa} \mid \boldsymbol{X}_t^{aa})$, to compute the distribution for each residue $i$: $p_\theta(\boldsymbol{x}_{t-1}^i \mid \boldsymbol{x}_t^i) \propto \sum_{\hat{\boldsymbol{x}}_0^i} q(\boldsymbol{x}_{t-1}^i \mid \boldsymbol{x}_t^i, \hat{\boldsymbol{x}}_0^i)\hat{p}_\theta(\hat{\boldsymbol{x}}_0^i \mid \boldsymbol{x}_t^i)$, where $\hat{\boldsymbol{x}}_0^i$ represents the predicted probability distribution for the $i$-th residue of the original sequence. The posterior distribution $q(\boldsymbol{x}_{t-1}^i \mid \boldsymbol{x}_t^i, \hat{\boldsymbol{x}}_0^i)$ can be calculated in closed form using the forward process transition matrices from Bayes' theorem:

$$q(\boldsymbol{x}_{t-1}^i \mid \boldsymbol{x}_t^i, \hat{\boldsymbol{x}}_0^i) = \frac{q(\boldsymbol{x}_t^i \mid \boldsymbol{x}_{t-1}^i, \hat{\boldsymbol{x}}_0^i)q(\boldsymbol{x}_{t-1}^i \mid \hat{\boldsymbol{x}}_0^i)}{q(\boldsymbol{x}_t^i \mid \hat{\boldsymbol{x}}_0^i)} = \text{Cat}\left(\boldsymbol{x}_{t-1}^i; p = \frac{\boldsymbol{x}_t^i \boldsymbol{Q}_t^\top \odot \hat{\boldsymbol{x}}_0^i \bar{\boldsymbol{Q}}_{t-1}}{\hat{\boldsymbol{x}}_0^i \bar{\boldsymbol{Q}}_t \boldsymbol{x}_t^{i\top}}\right), \quad (1)$$

where $\text{Cat}(\boldsymbol{x}; p)$ is a categorical distribution over $\boldsymbol{x}$ with probabilities determined by $p$. Assuming independence between residues conditioned on the previous state, the probability for the entire sequence is the product of the individual amino acid probabilities: $p_\theta(\boldsymbol{X}_{t-1}^{aa} \mid \boldsymbol{X}_t^{aa}) = \prod_{1 \le i \le N} p_\theta(\boldsymbol{x}_{t-1}^i \mid \boldsymbol{x}_t^i)$. To generate a complete new sequence, the process begins with a random noise sequence sampled from the prior distribution, $\boldsymbol{X}_T^{aa}$. This sequence is then iteratively denoised at each timestep using the reverse denoising process, eventually converging to a desired sequence $\boldsymbol{X}_0^{aa}$. To accelerate this iterative generation, we also employ a discrete denoising diffusion implicit model (DDIM) (Song et al., 2020) sampler (Bai et al., 2025).

## 3.2 PROBLEM FORMULATION: A NEW PERSPECTIVE

The goal of protein inverse folding is to predict the amino acid sequence based on the 3D conformation ($\mathcal{G}$). Unlike existing methods that only use 3D conformation as input, our method leverages existing protein data to enrich the feature.

**3D conformation.** We represent the protein structure as a graph $\mathcal{G} = (\mathcal{V}, \mathcal{E})$, where each node $v_i \in \mathcal{V}$ corresponds to an amino acid. The graph's connectivity is defined using a k-nearest neighbor (kNN) algorithm constrained by a distance cutoff. In particular, an edge $e_{ij}$ exists between two

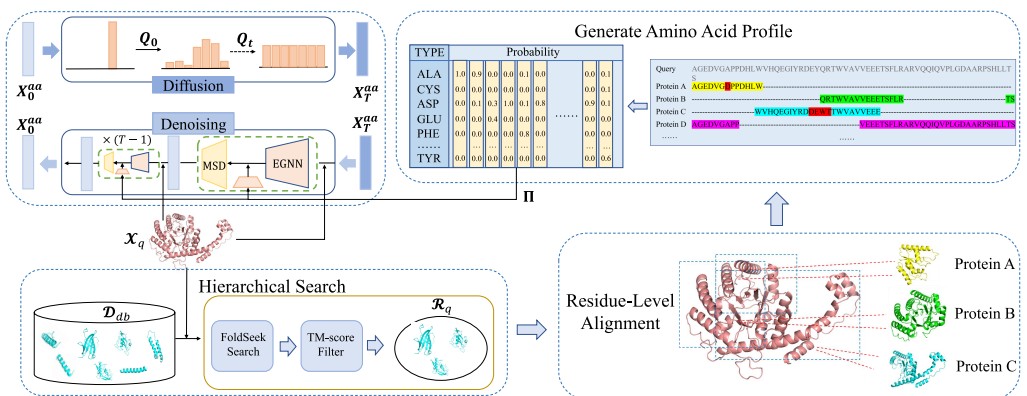

Figure 1: RadDiff's pipeline. (1) Hierarchical search: retrieve a structurally similar protein set $\mathcal{R}_q$ from the database $\mathcal{D}_{db}$. (2) Residue-level alignment: superimpose the retrieved proteins onto the query structure using US-align, and use the aligned residues as references for the amino acid types in the original sequence. (3) Generating amino acid profile: the amino acid profile is the position-specific amino acid probabilities, which serves as an evolutionary-informed prior that directly conditions the denoising process. The red color denotes the incorrect amino acid type after aligning, while the other color denotes the correct type.

nodes only if their $C_\alpha$ distance is less than 30Å. The input to the network consists of node features $\boldsymbol{H}$, coordinate features $\boldsymbol{X}^{pos}$, and edge features $\boldsymbol{A}$, which encode the geometric and relational properties of the structure. More details of the constructed features are described in Appendix A.1.

**Structure retrieval dataset.** Let protein $\mathcal{P}$ denotes the pair of amino acid sequence $\mathcal{S}$ and its 3D backbone structure $\mathcal{X}$, such that $\mathcal{P} = (\mathcal{S}, \mathcal{X})$. The protein inverse folding problem aims to find a valid sequence $\mathcal{S}$ that folds into the desired structure $\mathcal{X}$. Let $\mathcal{D}_{db} = \{\mathcal{P}'_j = (\mathcal{S}'_j, \mathcal{X}'_j)\}_{j=1}^M$ be an external database of $M$ known protein sequences and their structures. Given a query structure $\mathcal{X}_q$ with an unknown sequence, we first introduce a retrieval step to find a set of structurally similar proteins from $\mathcal{D}_{db}$. The retrieved set is denoted as $\mathcal{R}_q$, where

$$\mathcal{R}_q = \{\mathcal{P}'_1, \mathcal{P}'_2, ..., \mathcal{P}'_k\} \subset \mathcal{D}_{db}. \tag{2}$$

Here, $\mathcal{R}_q$ contains the most structurally similar proteins to $\mathcal{X}_q$ based on the designed similarity measurement.

The learning objective is to find a function $\mathcal{F}_\theta$ that models a conditional probability distribution, which depends not only on the target structure but also on the retrieved set $\mathcal{R}_q$. The model is trained to predict the sequence $\mathcal{S}_q$ given both $\mathcal{G}$ and the retrieved evidence $\mathcal{R}_q$:

$$\mathcal{F}_\theta : (\mathcal{G}, \mathcal{R}_q) \rightarrow \mathcal{S}_q. \tag{3}$$

### 3.3 Retrieval Augmentation

Our method conditions the protein inverse folding process on a prior derived from structurally similar proteins. This is achieved through the following three stages as shown in Figure 1: a hierarchical search for candidate structures, a precise residue-level alignment, and the generation of a position-specific amino acid profile.

### 3.3.1 Hierarchical Search

Given the vast size of protein databases, we design a hierarchical search strategy to efficiently identify a set of proteins that share similar structure with the query structure $\mathcal{X}_q$.

First, for coarse-grained filtering, we use FoldSeek (van Kempen et al., 2022) (Section 2) to perform a rapid search of $\mathcal{X}_q$ against the entire database $\mathcal{D}_{db}$. FoldSeek represents 3D structures as sequences of discrete structural alphabet identifiers (3Di). We leverage this representation to perform an initial filtering based on fident, defined as the fraction of identical 3Di characters in the alignment between two structures. We retain only those proteins with a fident score greater than 0.5. This process yields an initial candidate set, $\mathcal{D}_q \subset \mathcal{D}_{db}$, significantly reducing the search space for the next stage.

Second, for fine-grained filtering, we further refine $\mathcal{D}_q$ using US-align (Zhang & Skolnick, 2005) (Section 2), which performs coordinate-based structural alignment and calculates the TM-score. As the TM-score is asymmetric (its value depends on the reference protein length), an alignment between two proteins produces two scores, $tm_1$, $tm_2$. To specifically identify local structural matches—where a smaller protein may align perfectly to a fragment of a larger one—we retain all structures where $\min(tm_1, tm_2) > 0.5$. This ensures that even partial, high-quality fragment matches are preserved. The final set of $k$ retrieved proteins, denoted as $\mathcal{R}_q$, is obtained through this process. Both Foldseek and US-align are sequence-independent, ensuring that this retrieval process is based solely on structural information $\mathcal{X}$.

### 3.3.2 RESIDUE-LEVEL ALIGNMENT

After retrieving the set $\mathcal{R}_q$, we establish a precise residue-wise correspondence between the query structure $\mathcal{X}_q$ and each retrieved structure $\mathcal{X}_r$ from $\mathcal{P}_r = (\mathcal{S}_r, \mathcal{X}_r) \in \mathcal{R}_q$. The US-align algorithm, used in the previous step, provides this alignment as a byproduct of its structural superposition calculation.

As shown in Figure 1, for each retrieved protein $\mathcal{P}_r$, the alignment produces a mapping between residues in the query and residues in the retrieved structures. For each residue position $i$ in the query $\mathcal{P}_q$, the alignment either identifies a corresponding residue $j$ in $\mathcal{P}_r$ or indicates that position $i$ is not aligned to any residue.

We aggregate this information across all $k$ proteins in $\mathcal{R}_q$. For each position $i$ in the query sequence, we construct a multiset $\mathcal{T}_i$, containing the amino acid types of all aligned residues from the retrieved set:

$$\mathcal{T}_i = \{\mathcal{S}_r[j] \mid \forall \mathcal{P}_r = (\mathcal{S}_r, \mathcal{X}_r) \in \mathcal{R}_q \text{ where residue } i \text{ of } \mathcal{X}_q \text{ aligns with residue } j \text{ of } \mathcal{X}_r\}. \quad (4)$$

This multiset $\mathcal{T}_i$ serves as a collection of observed amino acid types that are evolutionarily and structurally compatible with the local backbone environment at position $i$.

### 3.3.3 GENERATE AMINO ACID PROFILE

From the collected multisets of amino acids $\{\mathcal{T}_i\}_{i=1}^N$, we generate a position-specific probability matrix, or namely amino acid profile $\mathbf{\Pi} \in \mathbb{R}^{N \times |\mathcal{V}|}$, where $|\mathcal{V}| = 20$ is the size of the amino acid vocabulary. This profile quantifies the preference for each amino acid type at each position.

The profile is calculated as a position-wise frequency distribution. For the $i$-th residue and amino acid type $aa \in \mathcal{V}$, and the profile value $\mathbf{\Pi} \in \mathbb{R}_{i,aa}$ is computed as:

$$\mathbf{\Pi}_{i,aa} = \begin{cases} \frac{\text{count}(aa \in \mathcal{T}_i)}{|\mathcal{T}_i|} & \text{if } |\mathcal{T}_i| > 0 \\ \frac{1}{|\mathcal{V}|} & \text{if } |\mathcal{T}_i| = 0 \end{cases}, \quad (5)$$

where $\text{count}(aa \in \mathcal{T}_i)$ is the number of times amino acid $aa$ appears in the multiset $\mathcal{T}_i$. For the unaligned positions (i.e., where no similar structure is retrieved or no residue on retrieved structure aligns with a particular residue $i$), the multiset $\mathcal{T}_i$ will be empty. For these positions, we assign a uniform distribution to provide a non-informative prior.

The resulting profile $\{\mathbf{\Pi}_i\}_{i=1}^N$ serves as an evolutionary-informed prior. This profile is then used as an additional conditioning input to our denoising model, guiding the sequence generation towards amino acid choices validated in known structures.

### 3.4 3D CONFORMATION REPRESENTATION

To represent 3D protein structure, we employ a global-aware equivariant graph neural network (EGNN) (Satorras et al., 2021) as the network backbone. The EGNN is composed of $L$ layers, where the $l$-th layer updates the node features $\boldsymbol{h}_i^l$ and coordinates $\boldsymbol{x}_i^l$ while preserving SE(3) equivariance. The $\boldsymbol{x}_i^0$ is $\boldsymbol{X}_i^{\text{pos}}$. At the $l$-th layer, the coordinates and node features are updated via:

$$\boldsymbol{m}_{ij}^l = \phi_e\left(\boldsymbol{h}_i^l, \boldsymbol{h}_j^l, \left\|\boldsymbol{x}_i^l - \boldsymbol{x}_j^l\right\|^2, \boldsymbol{a}_{ij}\right), \qquad \boldsymbol{x}_i^{l+1} = \boldsymbol{x}_i^l + \frac{1}{|\mathcal{N}_i|} \sum_{j \in \mathcal{N}_i} \left(\boldsymbol{x}_i^l - \boldsymbol{x}_j^l\right) \phi_x\left(\boldsymbol{m}_{ij}^l\right),$$

$$\boldsymbol{m}_i^l = \sum_{j \in \mathcal{N}_i} w_{ij} \boldsymbol{m}_{ij}^l, \qquad\qquad\qquad \boldsymbol{h}_i^{l+1} = \phi_h\left(\boldsymbol{h}_i^l, \boldsymbol{m}_i^l\right), \quad (6)$$

where $\mathcal{N}_i$ is the set of neighbors of node $i$ and $\phi_e, \phi_x, \phi_h$ are multi-layer perceptrons (MLPs). $\boldsymbol{a}_{ij}$ is the edge feature between node $i$ and $j$. $\boldsymbol{w}_{ij} = \sigma(\phi_w(\boldsymbol{a}_{ij}))$, and $\sigma(\cdot)$ is the sigmoid function.

We enhance this local message passing with a global context vector to allow for long-range communication across the structure (Tan et al., 2023; Bai et al., 2025). After the local update, the node representations are further refined as:

$$\boldsymbol{c}^{l+1} = \text{MeanPool}(\{\boldsymbol{h}_i^{l+1}\}_{i=0}^{N-1}), \qquad \boldsymbol{h}_i^{l+1} = \boldsymbol{h}_i^{l+1} \odot \sigma(\phi_c(\boldsymbol{c}^{l+1}, \boldsymbol{h}_i^{l+1})), \tag{7}$$

where $\odot$ is the Hadamard product. The above updating process is repeated for $L$ times. Finally, EGNN yields $\boldsymbol{h}_i^L$, the embedding of the final layer, as a representation of residue $i$.

## 3.5 EVOLUTIONARY-INFORMED GUIDING

We design two evolutionary-informed guiding approaches for the generation process, which is the integration of the amino acid profile and masked-prior-guided denoising (Bai et al., 2025).

**Integrate Amino Acid Profile**  The retrieval-based amino acid profile ($\{\boldsymbol{\Pi}_i\}_{i=1}^N$, Section 3.3), serving as an evolutionary-informed prior, is combined with 3D conformation representations ($\{\boldsymbol{h}_i^L\}_{i=1}^N$, Section 3.4). Integration is achieved via an extremely lightweight fusion module. The profile vector $\boldsymbol{\Pi}_i$ is first projected into the hidden dimension of the node features and then fused with $\boldsymbol{h}_i^L$ using a residual connection. The resulting feature is further refined to produce the final probability over amino acid categories:

$$\boldsymbol{p}_i = \text{softmax}(\phi_2(\phi_1(\boldsymbol{\Pi}_i) + \boldsymbol{h}_i^L)), \tag{8}$$

where $\phi_1$ and $\phi_2$ are MLPs. The network is trained to minimize the cross-entropy loss between $\boldsymbol{p}_i$ and the ground truth sequence.

**Masked-Prior-Guided Denoising**  Following MapDiff (Bai et al., 2025), to incorporate prior knowledge of sequential context, we pre-train a separate masked sequence designer (MSD). The role of the masked sequence design is to refine the residues with low predicting confidence. However, unlike MapDiff, our MSD is pre-trained independently and does not integrate into the training process of the graph denoising model. We found that this modification will accelerate the training process and can still improve the quality of the predicted sequence. The invariant point attention (IPA) network, first proposed by AlphaFold2 (Jumper et al., 2021) and modified by Frame2seq (Akpinaroglu et al., 2023) to incorporate geometric information, is used as the backbone of MSD. In the training stage, the MSD is trained based on the masked language modeling objective proposed in Devlin et al. (2019). MSD takes the masked amino acid sequence and the backbone coordinates as the input, and is trained to predict the original amino acid type using a cross-entropy loss. In the inference stage, given the output probability distribution $\boldsymbol{p}_i$ of the graph diffusion model, the entropy of the residue $i$ is defined as:

$$\text{entropy}_i = -\sum_j \boldsymbol{p}_{ij} \log(\boldsymbol{p}_{ij}). \tag{9}$$

The lowest entropy residues are masked during inference and predicted by the MSD. The detailed process of re-predicting the amino acid type is provided in Appendix A.2.

## 4 EXPERIMENT

### 4.1 EVALUATION SETTINGS

**RAG Database and Leakage Prevention.**  Wwe utilize the AlphaFold predicted (Varadi et al., 2022) Swiss-Prot (Bairoch & Apweiler, 1997) database, which contains 542,380 protein structures, as the external protein database. To prevent data leakage and ensure that our model's performance is not inflated by trivial matches, we implement a strict filtering protocol during the retrieval process for each query as shown in Figure 2. (a) Identity filtering: Any structure in the retrieval database that is identical to a structure within the test sets is excluded from the retrieval pool. (b) Substring filtering: For domain-based datasets like CATH, where test samples may be fragments of full proteins, we perform a sequence-based check. If the amino acid sequence of a potential database hit contains the query's domain sequence as a substring, or if the query's sequence contains the database sequence

as a substring, that hit is discarded. This filtering strategy ensures that the structural augmentation is derived from truly homologous structures rather than from artifacts of dataset construction, thereby providing a true evaluation of our model's ability to generalize.

**Baselines.** We use the following three categories of baseline methods for comparison: (1) GNN-based methods, including AlphaDesign (Gao et al., 2022a), ProteinMPNN (Dauparas et al., 2022), StructGNN (Ingraham et al., 2019), Graph-Trans (Ingraham et al., 2019), GVP (Jing et al., 2020), and PiFold (Gao et al., 2022b); (2) PLM-based methods, including LM-Design (Zheng et al., 2023) and KW-design (Gao et al., 2023); (3) diffusion-based methods, including GradeIf (Yi et al., 2023) and MapDiff (Bai et al., 2025). All baselines are evaluated under identical experimental settings.

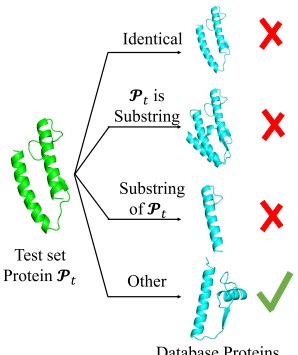

Figure 2: Illustration of strategy to prevent data leakage.

**Datasets.** Our evaluation employs several standard benchmark datasets to assess model performance thoroughly. Our main evaluation is conducted on the widely-used CATH v4.2 and v4.3 datasets (Orengo et al., 1997). Following established protocols, we adapt a topology-based data split to prevent overlap between the training, validation, and test sets. For CATH v4.2, the dataset is partitioned into 18,024 training, 608 validation, and 1,120 test samples. Similarly, the CATH v4.3 dataset is split into 16,630 training, 1,516 validation, and 1,864 test samples. To assess the zero-shot generalization capabilities of all models, we evaluate on two independent test sets. TS50 (Li et al., 2014) is a common benchmark containing 50 diverse protein chains. The PDB2022 dataset, curated by (Zhou et al., 2023), consists of 1,975 structures published in the Protein Data Bank (PDB) (Berman et al., 2000) between January 5, 2022, and October 26, 2022. This provides a strict, time-based split for evaluating temporal generalization. Both datasets are entirely separate from the CATH-derived training set, minimizing data leakage and providing a robust evaluation of structural and temporal generalization (Bai et al., 2025).

## 4.2 PROTEIN DESIGN ON CATH

**Accuracies**. The performance of RadDiff on the CATH v4.2 and CATH v4.3 datasets is summarized in Table 1. We use the same evaluation protocol as prior work Zheng et al. (2023); Gao et al. (2023); Bai et al. (2025), and the results of baselines are copied from the original papers. RadDiff achieves a new state-of-the-art, outperforming all baseline methods in both perplexity and sequence recovery rate across the short, single-chain, and full dataset splits. Specifically, on the CATH v4.2 benchmark, RadDiff achieves a perplexity of 2.46 and a sequence recovery of 67.14%. This represents a 28.9% reduction in perplexity and a 10.01% relative improvement in recovery rate over the previous best method. RadDiff also demonstrates similarly strong performance on the CATH v4.3 dataset, achieving a 39.0% reduction in perplexity and a 19.0% improvement in sequence recovery over the previous best method. Overall, the experimental results show the effectiveness of RadDiff.

**Model Size.** A key advantage of RadDiff is its parameter efficiency. While PLM-based methods like LM-Design and KW-Design also leverage external protein knowledge, they do so with substantial parameter overhead, requiring $46\times$ and $56\times$ more parameters than RadDiff, respectively. In contrast, RadDiff successfully integrates this external knowledge with a minimal increase in model size. These results show that RadDiff is a parameter-efficient way to use external database.

**Run Time.** The total retrieval augmentation process is computationally efficient. As detailed in Appendix B.2, RadDiff only requires a couple of minutes to compare 600 million structure pairs and obtain the alignment, so RadDiff is easily adaptable to the ever-growing protein database.

## 4.3 ZERO-SHOT GENERALIZATION ON PDB AND TS50

To assess the model's generalization capabilities, we evaluate its zero-shot performance on two independent datasets, TS50 and a new PDB dataset, using models trained on CATH v4.2 and CATH v4.3. As detailed in Table 2, the evaluation metrics includes median recovery rate and native sequence similarity recovery (NSSR) (Löffler et al., 2017). NSSR measures the biochemical similarity

Table 1: Performance on CATH v4.2 and CATH v4.3 datasets.

| Models | Model Size | Perplexity (↓) | | | Median Recovery Rate (%, ↑) | | |
|---|---|---|---|---|---|---|---|
| | | Short | Single-chain | Full | Short | Single-chain | Full |
| **CATH v4.2** | | | | | | | |
| StructGNN (Ingraham et al., 2019) | 1.4M | 8.29 | 8.74 | 6.40 | 29.44 | 28.26 | 35.91 |
| GraphTrans (Ingraham et al., 2019) | 1.5M | 8.39 | 8.83 | 6.63 | 28.14 | 28.46 | 35.82 |
| GVP (Jing et al., 2020) | 2.0M | 7.09 | 7.49 | 6.05 | 32.62 | 31.10 | 37.64 |
| AlphaDesign (Gao et al., 2022a) | 6.6M | 7.32 | 7.63 | 6.30 | 34.16 | 32.66 | 41.31 |
| ProteinMPNN (Dauparas et al., 2022) | 1.9M | 6.90 | 7.03 | 4.70 | 36.45 | 35.29 | 48.63 |
| PiFold (Gao et al., 2022b) | 6.6M | 5.97 | 6.13 | 4.61 | 39.17 | 42.43 | 51.40 |
| LM-Design (Zheng et al., 2023) | 659M | 6.86 | 6.82 | 4.55 | 37.66 | 38.94 | 53.19 |
| KW-Design (Gao et al., 2023) | 798M | 5.48 | 5.16 | 3.46 | 44.66 | 45.45 | 60.77 |
| GradeIf (Yi et al., 2023) | 7.0M | 5.65 | 6.46 | 4.40 | 45.84 | 42.73 | 52.63 |
| MapDiff (Bai et al., 2025) | 14.1M | 3.99 | 4.43 | 3.46 | 52.85 | 50.00 | 61.03 |
| RadDiff | 14.2M | **2.97** | **2.55** | **2.46** | **63.37** | **66.73** | **67.14** |
| **CATH v4.3** | | | | | | | |
| GVP-GNN-Large (Hsu et al., 2022) | 21M | 7.68 | 6.12 | 6.17 | 32.60 | 39.40 | 39.20 |
| ProteinMPNN (Dauparas et al., 2022) | 1.9M | 6.12 | 6.18 | 4.63 | 40.00 | 39.13 | 47.66 |
| PiFold (Gao et al., 2022b) | 6.6M | 5.52 | 5.00 | 4.38 | 43.06 | 45.54 | 51.45 |
| LM-Design (Zheng et al., 2023) | 659M | 6.01 | 5.73 | 4.47 | 44.44 | 45.31 | 53.66 |
| KW-Design (Gao et al., 2023) | 798M | 5.47 | 5.23 | 3.49 | 43.86 | 45.95 | 60.38 |
| GradeIf (Yi et al., 2023) | 7.0M | 5.30 | 6.05 | 4.58 | 48.21 | 45.94 | 52.24 |
| MapDiff (Bai et al., 2025) | 14.1M | 3.88 | 3.85 | 3.48 | 55.95 | 54.65 | 60.86 |
| RadDiff | 14.2M | **2.48** | **2.35** | **2.38** | **75.62** | **75.00** | **72.40** |

Table 2: Generalizability evaluation on PDB2022 and TS50 datasets. The results in brackets are from the model trained with CATH v4.3.

| Models | PDB2022 | | | TS50 | | |
|---|---|---|---|---|---|---|
| | Recovery(↑) | NSSR62(↑) | NSSR90(↑) | Recovery(↑) | NSSR62(↑) | NSSR90(↑) |
| ProteinMPNN (Dauparas et al., 2022) | 56.75 (56.65) | 72.50 (72.59) | 69.96 (69.95) | 52.34 (51.80) | 70.31 (70.13) | 66.77 (66.80) |
| PiFold (Gao et al., 2022b) | 60.63 (60.26) | 75.55 (75.30) | 72.96 (72.86) | 58.39 (58.90) | 73.55 (74.52) | 70.33 (71.33) |
| LM-Design (Zheng et al., 2023) | 66.03 (66.20) | 79.55 (80.12) | 77.60 (78.20) | 57.62 (58.27) | 73.74 (75.69) | 71.22 (73.12) |
| GradeIf (Yi et al., 2023) | 58.09 (58.35) | 77.44 (77.51) | 74.57 (74.97) | 57.74 (59.27) | 77.77 (79.11) | 74.36 (76.24) |
| MapDiff (Bai et al., 2025) | 68.03 (68.00) | 84.19 (84.30) | 82.13 (82.29) | 68.76 (69.77) | 84.10 (85.27) | 81.76 (83.08) |
| RadDiff | **76.22 (75.70)** | **87.38 (85.62)** | **86.37 (84.06)** | **75.64 (76.99)** | **88.98 (91.10)** | **86.91 (88.65)** |

between predicted and native residues using the BLOSUM (Henikoff & Henikoff, 1992) substitution matrix, where a residue pair is considered a match if its BLOSUM score is positive. NSSR62 and NSSR90 denote the use of the BLOSUM62 and BLOSUM90 matrices, respectively. We can find that RadDiff consistently outperforms all baselines in both recovery rate and NSSR, regardless of the training dataset. On the PDB dataset, models trained on CATH v4.2 and v4.3 achieve recovery rate of 76.22% and 75.70%, improving upon the previous best methods by 12.04% and 11.32%. On the TS50 dataset, the models achieve recovery rate of 75.64% and 76.99% respectively, representing 10.00% and 10.35% improvements over MapDiff. Furthermore, RadDiff obtains the highest NSSR62 and NSSR90 scores, demonstrating its superior ability to not only predict the correct amino acid but also to capture biochemically meaningful residue similarities. Overall, the results show that RadDiff generalizes well on unseen data.

## 4.4 ANALYSIS OF RETRIEVAL-AUGMENTED PERFORMANCE

**Impact of Retrieval Augmentation.** To investigate how the retrieved structures will influence the performance. We first quantify the retrieval coverage across the 1,120 proteins in the CATH v4.2 test set. As detailed in Table 4, 47.86% of proteins have at least one suitable hit ("w. RAG"), while the remaining 52.14% lack matches ("w.o. RAG"). As shown in Table 3, RadDiff achieves 89.80% recovery on the "w. RAG" subset, which is 31% higher than the "w.o. RAG" subset. This demonstrates that the retrieval-based amino acid profile serves as an effective prior for the generation process. We further investigate the granularity of the retrieval guidance at the residue level. Table 4 also reveals that within the "w. RAG" subset, 62.40% of the aligned residues are correctly matched. In contrasted with the 89.80% recovery rate achieved on this subset, it demonstrates that our model is not merely "copying" the retrieved amino acids. Instead, the model accurately infers the identities of the remaining 37.6% of unaligned positions with very high fidelity.

Table 3: Performance comparison on the CATH v4.2 test set classified by retrieval success.

| Metric | w. RAG | w.o. RAG |
|---|---|---|
| Recovery Rate (%)↑ | 89.80 | 58.64 |
| Perplexity↓ | 1.56 | 4.01 |

Table 4: Retrieval and alignment coverage statistics.

| | Number | Ratio (%) |
|---|---|---|
| Proteins w/ RAG Hit | 536 | 47.86 |
| Proteins w/o RAG Hit | 584 | 52.14 |
| Total Aligned Residues | 1.818M | 62.40 |
| Total Unaligned Residues | 1.095M | 37.60 |

Table 5: Foldability comparison using Boltz and ESMFold.

| Models | Boltz2 | | | | ESMFold | |
|---|---|---|---|---|---|---|
| | TMscore (↑) | RMSD (↓) | pTM (↑) | pLDDT (↑) | TMscore (↑) | RMSD (↓) |
| ProteinMPNN (Dauparas et al., 2022) | $84.95_{\pm 16.36}$ | $1.66_{\pm 0.94}$ | $82.66_{\pm 15.47}$ | $86.74_{\pm 10.73}$ | $84.34_{\pm 18.00}$ | $\mathbf{1.78}_{\pm \mathbf{1.03}}$ |
| PiFold (Gao et al., 2022b) | $84.77_{\pm 15.85}$ | $1.72_{\pm 0.90}$ | $82.48_{\pm 14.33}$ | $86.08_{\pm 10.04}$ | $81.82_{\pm 18.63}$ | $1.97_{\pm 1.10}$ |
| LM-Design (Zheng et al., 2023) | $83.98_{\pm 16.78}$ | $1.73_{\pm 0.94}$ | $83.11_{\pm 14.67}$ | $87.16_{\pm 9.69}$ | $80.87_{\pm 19.33}$ | $1.98_{\pm 1.13}$ |
| GradeIf (Yi et al., 2023) | $78.55_{\pm 17.46}$ | $2.27_{\pm 0.97}$ | $74.43_{\pm 15.04}$ | $78.50_{\pm 11.50}$ | $73.79_{\pm 20.94}$ | $2.58_{\pm 1.26}$ |
| MapDiff (Bai et al., 2025) | $84.71_{\pm 14.94}$ | $1.78_{\pm 0.83}$ | $82.32_{\pm 12.72}$ | $86.04_{\pm 8.86}$ | $82.03_{\pm 17.00}$ | $2.01_{\pm 1.00}$ |
| RadDiff | $\mathbf{87.69}_{\pm \mathbf{13.06}}$ | $\mathbf{1.55}_{\pm \mathbf{0.76}}$ | $\mathbf{85.58}_{\pm \mathbf{11.32}}$ | $\mathbf{89.70}_{\pm \mathbf{6.78}}$ | $\mathbf{85.43}_{\pm \mathbf{14.74}}$ | $1.79_{\pm 0.90}$ |

**Foldability of RAG-enabled Designs.** Beyond recovery rate, a critical measure of success for the design methods is the *in silico* foldability of the generated sequences. To assess this, we employ two cutting-edge structure prediction models, MSA-based method Boltz2 (Passaro et al., 2025) and MSA-free method ESMFold (Lin et al., 2022), to predict the tertiary structures of sequences. AlphaFold2 is not used due to its prohibitively expensive local MSA search process. The results in Table 5 focus specifically on the "w. RAG" subset of the test set, allowing for a direct comparison of design quality when our method is successfully guided by retrieved proteins. We compare the re-folded structures to the ground-truth crystal structures using a suite of metrics, including predicted TM-score (pTM), predicted aligned error (PAE), and predicted local distance difference test (pLDDT) from Boltz2, as well as the TM-score and RMSD from direct structural alignment. Across all metrics, RadDiff yields higher confidence scores and structural similarity to the native fold except the RMSD on ESMFold predicted structures. The results demonstrate that RadDiff produces designs that are highly likely to fold to the intended structure.

**Influence of Database Size.** To investigate how the size of the external database will influence the model performance, we conduct an experiment where we vary the size of the external database and measure the impact on both the retrieval hit numbers and the sequence recovery rate. As shown in Figure 3, there is a positive correlation between the database size and the number of test set queries for which similar structures are found. We can find that a larger database provides a greater opportunity to find similar structures, and this increase in retrieval coverage translates directly to improved model performance. The overall recovery rate for the entire test set rises with the database enlarged from 10,000 to 500,000, increasing from approximately 59% to over 67%. The result shows the scalability of RadDiff and the potential of increasing the database size to improve the performance.

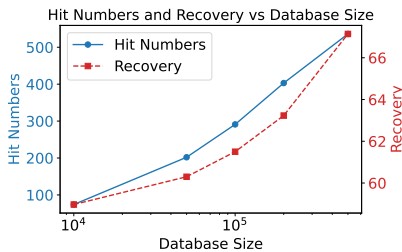

Figure 3: The relationship between the size of external database, hit numbers, and sequence recovery.

## 5 CONCLUSION

In this paper, we have introduced RadDiff, a novel retrieval-augmented denoising diffusion method for protein inverse folding. Experimental results on the CATH, PDB, and TS50 datasets show that RadDiff consistently outperforms existing methods, improving sequence recovery rate by up to 19%. Experimental results also demonstrate that RadDiff generates highly foldable sequences and scales effectively with database size.

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

# A  MORE METHOD DETAILS

## A.1  DETAIL OF GRAPH CONSTRUCTION

We will explain the detailed process of obtaining graph features of $\boldsymbol{H}, \boldsymbol{X}^{pos}, \boldsymbol{A}$. The node feature $\boldsymbol{H}$ contains several properties that describe the residue and its local environment: the residue type, secondary structure, dihedral angles, solvent-accessible surface area (SASA), crystallographic B-factor, and protein surface features (Ganea et al., 2021; Yi et al., 2023; Bai et al., 2025). The residue type is a one-hot encoding of the amino acid's type. The secondary structure is a one-hot encoding of the secondary structure element (e.g., helix, sheet, coil) assigned by DSSP (Kabsch & Sander, 1983). The dihedral feature is calculated as $\{sin, cos\} \circ \{\phi, \psi\}$. The SASA is a scalar value indicating the residue's exposure to the solvent. The crystallographic B-factor is a scalar value indicating the residue's mobility. The surface feature is defined as:

$$\rho_i(\boldsymbol{x}_i; \lambda) = \frac{\left\| \sum_{j \in \mathcal{N}_i} w_{i,j,\lambda} \left( \boldsymbol{x}_i^{\mathrm{pos}} - \boldsymbol{x}_j^{\mathrm{pos}} \right) \right\|}{\sum_{j \in \mathcal{N}_i} w_{i,j,\lambda} \left\| \boldsymbol{x}_i^{\mathrm{pos}} - \boldsymbol{x}_j^{\mathrm{pos}} \right\|}, \quad \text{where} \quad w_{i,j,\lambda} = \frac{\exp\left( -\|\boldsymbol{x}_i^{\mathrm{pos}} - \boldsymbol{x}_j^{\mathrm{pos}}\|^2 / \lambda \right)}{\sum_{j \in \mathcal{N}_i} \exp\left( -\|\boldsymbol{x}_i^{\mathrm{pos}} - \boldsymbol{x}_j^{\mathrm{pos}}\|^2 / \lambda \right)}, \tag{10}$$

where $\mathcal{N}_i$ is the set of neighboring nodes, and this calculation is performed for multiple scales with $\lambda \in \{1, 2, 5, 10, 30\}$.

The $\boldsymbol{X}^{\mathrm{pos}}$ is the coordinates of $C_\alpha$ atoms for each residue.

The Edge feature $\boldsymbol{A}$ contains the relative spatial distance, local spatial positions and relative sequential positions (Bai et al., 2025). The relative spatial distance feature encode the distance using Radial Basis Function (RBF). An additional binary contact signal is included, which is set to 1 if the spatial distance between the two residues is less than 8 Å, and 0 otherwise. The local spatial position derived from a local coordinate system constructed at each residue and represents the relative positions and orientations of the backbone atoms. The relative sequential positions encodes the separation of the two residues along the primary sequence. It is a one-hot encoding based on the difference in their sequential indices. The above three separate features are concatenated into the final edge features $\boldsymbol{A}$.

## A.2  DETAIL OF MASK SEQUENCE DESIGNER

The purpose of the masked sequence designer is to learn the conditional probability distribution of amino acids given their structural and sequential context. This allows it to refine low-confidence predictions made by the main denoising network during inference. Following the pre-training strategy of MapDiff, we use a masked language modeling objective. For each training sequence, a portion of the amino acid (AA) residues is randomly selected for corruption: 80% are replaced with a special [MASK] token, 10% are replaced with a random AA, and the remaining 10% are left unchanged. To prevent information leakage from external sources, the designer is pre-trained on the same CATH v4.2/v4.3 training sets used for the main diffusion model.

We employ an Invariant Point Attention (IPA) network as the architecture for the masked sequence designer. IPA is a geometry-aware attention mechanism, originally developed for AlphaFold2 (Jumper et al., 2021), designed to effectively fuse residue representations with their spatial relationships in 3D space. To incorporate the protein's 3D geometry, the IPA network operates on three key inputs derived from the query structure $\mathcal{X}$ and the masked sequence: (1) A feature matrix $\boldsymbol{R} \in \mathbb{R}^{n \times d_s}$, where $n$ is the sequence length. for each residue $i$, the feature vertor $\boldsymbol{R}_i$ is derived from the its AA type from the masked sequence and a positional encoding. (2) A feature tensor $\boldsymbol{Z} \in \mathbb{R}^{n \times n \times d_z}$ that encodes relational information between all pairs of residues. The feature $\boldsymbol{Z}_{ij}$ include the relative positions of residues $i$ and $j$ along the sequence and their spatial distance in 3D. (3) A set of rigid coordinate frames $\mathcal{T} = \{\boldsymbol{T}_i\}_{i=1}^n$, where each from $\boldsymbol{T}_i = (\boldsymbol{Rot}_i \in \mathbb{R}^{3 \times 3}, \boldsymbol{t}_i \in \mathbb{R}^3)$ consists of a rotation matrix and a translation vector. These frames are constructed from the backbone atom coordinates using the Gram-Schmidt process. The local frames ensure invariance of IPA to global Euclidean transformations. The input $(\boldsymbol{R}, \boldsymbol{Z}, \mathcal{T})$ are processed through a stack of $L$ IPA layers:

$$\boldsymbol{R}^{l+1}, \boldsymbol{Z}^{l+1} = \mathrm{IPA}(\boldsymbol{R}^l, \boldsymbol{Z}^l, \mathcal{T}). \tag{11}$$

After the final layer, the output residue representation $\boldsymbol{R}_i^L$ for each position $i$ is projected through a linear layer to produce logits, which are then converted into a probability distribution over the 20

amino acids via a softmax function:

$$z_i^m = \mathrm{Linear}(R_i^L) \tag{12}$$

$$p_i^m = \mathrm{softmax}(z_i^m), \tag{13}$$

$$\mathrm{entropy}^m(i) = -\sum_j p_{ij}^m \log(p_{ij}^m). \tag{14}$$

During inference, the final predicted probability is refined as:

$$p_i^f = \mathrm{softmax}(\frac{\exp(-\mathrm{ent}_i)}{\exp(-\mathrm{ent}_i) + \exp(-\mathrm{ent}_i^m)}z_i + \frac{\exp(-\mathrm{ent}_i^m)}{\exp(-\mathrm{ent}_i) + \exp(-\mathrm{ent}_i^m)}z_i^m), \tag{15}$$

where $z_i$ and $\mathrm{ent}_i$ is defined in equation 8 and 9.

### A.3 DETAIL OF DDIM

We use denoising diffusion implicit model (DDIM) (Song et al., 2020) to accelerate the denoising process. DDIM constructs a non-Markovian diffusion processes so that the sampling in reverse process can be faster. Follow the settings of Yi et al. (2023), the multi-step generative process is defined as:

$$p_\theta(x_{t-k}^i \mid x_t^i) \propto (\sum_{\hat{x}_0^i} q(x_{t-k}^i \mid x_t^i, \hat{x}_0^i)\hat{p}_\theta(\hat{x}_0^i \mid x_t^i))^T, \tag{16}$$

where $k$ is the number of skiping steps and $T$ controls whether it is deterministic or stochastic. The posterior distribution is:

$$q(x_{t-k}^i \mid x_t^i, \hat{x}_0^i) = \mathrm{Cat}\left(x_{t-k}^i; p = \frac{x_t^i Q_t^\top \cdots Q_{t-k}^\top \odot \hat{x}_0^i \bar{Q}_{t-k}}{\hat{x}_0^i \bar{Q}_t x_t^{i\top}}\right), \tag{17}$$

## B    MORE EXPERIMENTAL DETAILS

### B.1    IMPLEMENTATION DETAILS

The denoising network's backbone consists of EGNN with 6 layers, each with a hidden dimension of 128. the masked sequence designer is composed of 6 IPA layers, also with a hidden dimension of 128. We employ the Adam optimizer with an initial learning rate of $5 \times 10^{-4}$, managed by a one-cycle learning rate scheduler. A batch size of 8 is used for all training stages. Following the protocol of Bai et al. (2025), the MSD module is pre-trained for 200 epochs. The main graph-based denoising model is trained for 100 epochs. For the RAG process, FoldSeek (van Kempen et al., 2022) of version `9.427df8a` is installed. The database proteins is first encoded using the `foldseek createdb` command, and the initial rapid search is performed with the `foldseek easy-search` command. For the *in silico* foldability analysis, we use Boltz v2.03 (Passaro et al., 2025) to predict the 3D structures of the generated sequences. The MSAs required as input for Boltz are generated using the online MSA server.

### B.2    RUN TIME EVALUATION

**Retrieval Time**    A critical factor for the practical application of RadDiff is the computational efficiency of its retrieval process. To quantify this, we calculate the runtime of our hierarchical search strategy for all 1,120 query proteins in the CATH v4.2 test set against the Swiss-Prot database, which contains 542,380 structures. This corresponds to a total search space of over 600 million potential pairwise comparisons. The results, detailed in Table 6, demonstrate the high efficiency of our approach. The entire retrieval process for all 1,120 queries completed in just 306.5 seconds, corresponding to an average of only 0.27 seconds per query. The initial rapid filtering with FoldSeek required only 54.0 seconds in total (an average of 0.04s per query) to drastically narrow down the search space from the entire database. Subsequently, the more computationally intensive US-align step, which provides the high-quality alignments essential for our method, was applied only to this pre-filtered small set of candidates. The results demonstrates that our RAG approach is not only effective but also computationally practical.

Table 6: Retrieval Time

|        | FoldSeek | US-align | Total  |
|--------|----------|----------|--------|
| Time   | 53.98s   | 252.52s  | 306.5s |
| Time per Query | 0.04s | 0.23s | 0.27s |

**Protein Designing Time**   We also calculate the protein designing time of RadDiff, which cost around 13.45 seconds per sample, which is also practical in real-world applications.

### B.3   ABLATION STUDY

To systematically evaluate the contributions of the key components, We conduct ablation study on the RAG and the MSD module. We evaluate three distinct model configurations on the CATH v4.2 test set: (1) the full RadDiff model with both modules enabled, (2) a variant with only the RAG module, and (3) a variant with only the MSD module. The results is shown is Table 7. We can find that the RAG and MSD module both contribute to the final results. Specifically, the RAG module contribute to a sequence recovery improvement of 6.64%. and the MSD module also improve the sequence recovery by 4.13%. The ablation study demonstrate the effectiveness of our RAG and MSD module.

Table 7: Ablation study

| MSD | RAG | Perplexity↓ | Sequence Recovery(%)↑ |
|-----|-----|-------------|------------------------|
| ✓   | ✗   | 3.27        | 61.50                  |
| ✗   | ✓   | 3.39        | 63.03                  |
| ✓   | ✓   | **2.46**    | **67.14**              |

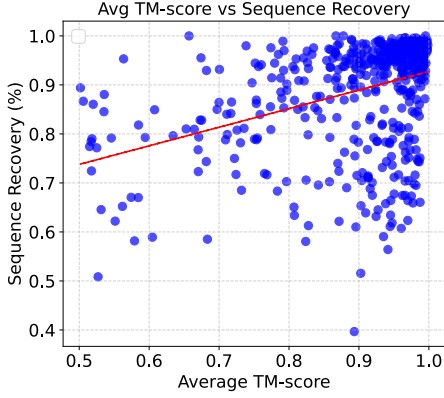

Figure 4: The relationship between the average TM-score of test proteins and their retrieved proteins, and sequence recovery.

**Influence of Retrieved Structural Similarity**   To understand how the quality of the retrieved set influences generation performance, we analyzed the relationship between structural similarity and sequence recovery. For each test sample in the RAG-enabled subset, we calculated the average TM-score across all of its retrieved structures and show this value against the sequence recovery. The results, shown in Figure 4, reveal two important insights. First, there is a positive trend, quantified by a Pearson correlation coefficient of 0.374. This indicates that, as expected, retrieving more structurally similar proteins generally leads to higher sequence recovery. The results also highlights the

robustness of RadDiff. Even when the average structural similarity of the retrieved set is modest (e.g., TM-score between 0.5 and 0.7), RadDiff consistently generates sequences with high recovery rates, often exceeding 70-80%.

## C  USAGE OF LANGUAGE MODELS

We utilized a large language model (LLM) to aid in the preparation of this manuscript. Its use was limited to editorial tasks, including proofreading for typographical errors, correcting grammar, and improving the clarity and readability of the text.

## D  REPRODUCIBILITY STATEMENT

We have taken several steps to ensure the reproducibility of our work. All datasets used in our experiments (CATH, PDB, and TS50) are publicly available, and we provide detailed descriptions of dataset splits and preprocessing procedures. We have provided part of the code. The full implementation of RadDiff, including the hierarchical search, residue-level alignment, generating amino acid profile, and denoising modules, will be released in an open-source repository upon publication. We also include training details, hyperparameter settings, and evaluation protocols to facilitate replication.

