# OpenReview forum: "RadDiff: Retrieval-Augmented Denoising Diffusion for Protein Inverse Folding"
_ICLR.cc/2026/Conference — ICLR 2026 Conference Withdrawn Submission_

### Official Review · Reviewer_1t3x · 2025-10-29

**Soundness:** 2
**Presentation:** 2
**Contribution:** 2
**Rating:** 4
**Confidence:** 4

**Summary:**

This manuscript introduces RadDiff, a retrieval-augmented denoising diffusion framework for protein inverse folding that aims to leverage external protein databases to improve sequence design. The core contribution lies in a hierarchical search strategy that efficiently retrieves structurally similar proteins from large databases, followed by a residue-level alignment mechanism that constructs position-specific amino acid profiles from retrieved structures. These profiles serve as evolutionary-informed priors integrated into a denoising diffusion model through a lightweight fusion module. The authors demonstrate that RadDiff achieves competitive performance on CATH, PDB, and TS50 benchmarks, with claimed improvements of up to 19% in sequence recovery rate over existing methods. Additionally, the authors provide evidence of computational efficiency and scalability with database size, along with favorable foldability metrics for generated sequences.

**Strengths:**

**S1**. While the core idea of leveraging structurally similar proteins through retrieval is not entirely novel in the protein design community, the authors have identified an important and promising research direction that addresses a fundamental limitation of current deep learning approaches. The motivation to incorporate evolutionary information stored in vast protein databases represents a scientifically sound approach. This direction aligns well with the growing recognition that external knowledge can substantially enhance generative models in specialized scientific domains.

**S2**. The manuscript demonstrates consistently strong empirical results across several established benchmarks, including CATH v4.2, CATH v4.3, PDB2022, and TS50 datasets. The reported improvements in sequence recovery rates and perplexity reductions compared to competitive baselines are substantial. Moreover, the authors provide comprehensive evaluation including not only sequence recovery but also structural metrics for foldability assessment, which demonstrates the thoroughness of the experimental validation and suggests that the generated sequences are both accurate and designable.

**Weaknesses:**

**W1**. The manuscript lacks crucial guidance and systematic analysis regarding the selection of $k$, the number of retrieved proteins, which appears to be a key hyperparameter in the proposed framework. The authors do not provide ablation studies showing how performance varies with different values of $k$, nor do they discuss the trade-offs between retrieval coverage and noise introduction from less relevant structures. Furthermore, there is no discussion of whether $k$ should be adapted based on query characteristics (e.g., protein family, structural complexity, or availability of similar structures in the database). This omission is particularly concerning given that the quality of the amino acid profile $\mathbf{\Pi}$ directly depends on $k$, and different proteins may benefit from different numbers of retrieved sequence templates.

**W2**. The manuscript does not provide critical analysis of the data distribution characteristics and potential overlap between training and test sets, which is essential for understanding the true generalization capabilities of the method. Specifically, there is no quantitative analysis of the structural similarity distribution between proteins in the training set and those in the test set. High overlap could lead to overly optimistic performance estimates, while understanding this overlap would help characterize when and why the method succeeds or fails. Additionally, analysis of the diversity within the retrieved sets for test queries would provide insight into whether the method truly leverages evolutionary information or primarily relies on near-duplicate retrieval.

**W3**. The evaluation protocol assumes that for each query structure, there exists a single ground truth sequence, which is fundamentally inconsistent with our understanding of protein structure-sequence relationships. It is well established in structural biology that multiple distinct sequences can fold into highly similar or nearly identical structures, such as hemoglobin. The paper does not address this many-to-one mapping between sequences and structures, raising concerns about whether low recovery rates might partially reflect valid alternative sequences rather than prediction errors.

**W4**. The experimental evaluation lacks several important baseline comparisons that would help isolate the contributions of different components and better position the work relative to simpler alternatives. First, a direct comparison with using only FoldSeek retrieval without the subsequent US-align refinement and denoising diffusion would clarify whether the complex pipeline is necessary or if a simple template-based approach is sufficient. Second, a baseline using the hierarchical search strategy but directly outputting the amino acid profile $\mathbf{\Pi}$ without the denoising diffusion model would isolate the contribution of the diffusion component versus the retrieval component.

**W5**. In Section 1, line 53, the authors claim that *incorporating new protein data from continuously growing protein databases requires retraining the entire PLM, which is both inflexible and computationally prohibitive*. This does not accurately reflect how pre-trained language models are typically used in practice. Pre-trained PLMs can generate representations for new protein sequences during inference without any retraining, which is precisely the purpose of pre-training and is analogous to how language models process new text. The actual limitation is not the inability to process new data, but rather that the model's learned statistical patterns are frozen at the pre-training cutoff date and do not automatically incorporate structural knowledge from newly discovered proteins.

**W6**. The current organization of the paper could be improved to enhance clarity and logical flow. Specifically, Sections 3.1 and 3.2 would be more appropriately positioned in Section 2 as background material, forming an expanded *Preliminaries* section. The current Section 2 and these subsections cover foundational concepts that should be established before presenting the method. This reorganization would allow Section 3 to begin directly with Section 3.3, making the paper's core contributions more immediately apparent and improving the overall organization of the manuscript.

**Questions:**

**Q1**. Could the authors provide a comprehensive ablation study on the effect of $k$ on model performance? Additionally, how sensitive is the model to other hyperparameters in the hierarchical search, such as the *fident* threshold (currently 0.5) used in FoldSeek and the TM-score threshold (currently 0.5) used in US-align?

**Q2**. Can the authors provide a detailed analysis of the TM-score distribution between test set proteins and their nearest neighbors in the training database? Specifically, what percentage of test proteins have retrieved structures with TM-scores in the ranges [0.5-0.6], [0.6-0.7], [0.7-0.8], [0.8-0.9], and [0.9-1.0]? How does model performance correlate with these similarity ranges?

**Q3**. Given that multiple sequences can fold into similar structures, have the authors considered evaluating whether their generated sequences, even when different from the target sequence, might still fold into the correct structure?

**Q4**. What is the individual contribution of: (i) FoldSeek-only retrieval without US-align refinement, (ii) using the amino acid profile $\mathbf{\Pi}$ directly without the diffusion model, and (iii) the masked sequence designer module? These ablations would help understand which components are essential for performance.

**Q5**. How does the method perform on proteins with genuinely novel folds that have no similar structures in the database? Does performance degrade gracefully *compared to other baselines*, or does the method fail catastrophically in such scenarios?

**Q6**. While the paper discusses scalability with database size, how does computational cost scale with protein length and structural complexity? Are there practical limitations for very large proteins or complex multi-domain structures?

**Details Of Ethics Concerns:**

No concerns

---

### Official Review · Reviewer_m4LB · 2025-10-30

**Soundness:** 2
**Presentation:** 4
**Contribution:** 1
**Rating:** 2
**Confidence:** 4

**Summary:**

The paper proposes a retrieval-augmented diffusion approach for protein inverse folding. Given a target backbone, the method (i) retrieves structurally similar proteins, (ii) aligns them at residue level, and (iii) converts the retrieved evidence into an amino-acid profile used to guide a discrete denoising diffusion model for sequence design. The system also includes a “Mask Sequence Designer (MSD)” refinement component to improve performance. Experiments are reported on CATH design and zero-shot tests on PDB/TS50, with ablations and runtime details summarized in the appendix.

**Strengths:**

1. The retrieval to alignment to profile-conditioning pipeline is a sensible way to inject evolutionary/structural signal into sequence design.
2. The manuscript is clearly structured (explicit sections on retrieval augmentation, hierarchical search, residue-level alignment, profile generation, and evolutionary guiding), and includes ablations/runtime/reproducibility notes.
3. The Mask Sequence Designer idea is potentially interesting as a refinement stage, and (if properly isolated) could be a useful contribution.

**Weaknesses:**

1. Limited novelty relative to prior practice

The central idea, retrieving close structural neighbors, aligning them, and converting to residue-wise priors, is well-established in the protein modeling literature (retrieval + alignment + MSA/profile conditioning). The paper’s main novelty appears to lie in integrating these established components within a diffusion framework plus an MSD refinement module. However, retrieval and alignment themselves employ existing tools and known procedures rather than new algorithms. The paper should more clearly articulate what is technically novel beyond combining prior ingredients.

Although the narrative emphasizes retrieval as a core contribution, this component is familiar. The only potentially novel aspect is the MSD, yet this is insufficiently explored: it is described briefly and evaluated in a single ablation, without deeper analysis. As a result, the work is framed around retrieval, which is not novel, while the genuinely new piece remains underdeveloped.

2. Leakage in retrieval and retrieval quality versus performance

Filtering solely on identical structures is too lenient; many studies use a TM-score threshold (e.g., 0.3) to define structural similarity. The authors briefly touch on this in the “impact of retrieval augmentation” section, but a systematic evaluation is missing. The paper should quantify how performance depends on retrieval quality:

Does excluding similar folds (low TM-score) degrade performance?

How does the number or quality of retrieved hits affect accuracy?

How does the MSA quality influence model behavior?

Because real-world protein design often targets novel structures with few close analogs, the current retrieval-dependent approach may fail in precisely the settings where inverse folding is most useful. This practical limitation should be analyzed empirically.

3. Positioning vs PLM-based baselines

The argument against PLM conditioning is unconvincing. Inverse-folding and design pipelines already incorporate PLM embeddings at inference without gradient flow through the model, an inexpensive yet powerful way to incorporate external evolutionary information. To claim novelty or superiority, the paper must compare against such PLM-augmented baselines under identical conditions. A simple test, injecting PLM embeddings in place of amino-acid residue frequencies, would clarify whether retrieval offers distinct benefits or simply provides another form of external knowledge.

4. Diversity is important

Beyond average recovery metrics, diversity is essential: an effective inverse-folding model should generate multiple diverse, high-quality designs. The paper does not assess diversity in sequence space (diversity that maintains good structural recovery). Moreover, retrieval conditioning could collapse diversity by biasing toward known templates. The authors must explore this.

5. (minor) Wording in abstract

The paper describes “experiments”; these are entirely computational. Please qualify as in-silico or “computational experiments” to avoid ambiguity. Given the context I initially assumed authors performed wet-lab experiments.

**Questions:**

See weakness

---

### Official Review · Reviewer_kCnh · 2025-11-01

**Soundness:** 2
**Presentation:** 2
**Contribution:** 1
**Rating:** 2
**Confidence:** 4

**Summary:**

This paper introduces RadDiff (Retrieval-Augmented Denoising Diffusion) to address the challenging task of protein inverse folding. The core methodology of RadDiff involves the dynamic retrieval of structural exemplars from large-scale protein databanks. This retrieved structural information is subsequently leveraged to guide the generative process of a denoising diffusion model. RadDiff circumvents the reliance on parameter-heavy protein language models, thereby striking a compelling balance among generative fidelity, parameter efficiency, and adaptability to newly incorporated data.

**Strengths:**

1. The motivation is well-grounded. Furthermore, the integration of dynamic structural information retrieval into the model's training paradigm represents a novel and compelling approach.
2. The proposed method outperforms baseline models across various metrics on benchmark datasets. Additionally, it boasts less parameters.

**Weaknesses:**

1. The claim that RadDiff adapts to new data without retraining lacks comprehensive assessment, as only a single database was used. Performance should be validated across diverse databases (e.g., PDB, AlphaFold DB) or different temporal versions.
2. Performance correlates with the number and quality (TM-score) of retrieved hits. An analysis of the trade-off between these two factors is preferred.
3. Given that both RadDiff and PRISM are retrieval-augmented methods, PRISM should be included as an experimental baseline for direct comparison.

**Questions:**

1. Is it better to retrieve few high-quality hits or many medium-quality ones?
2. The retrieval database uses AlphaFold predictions. How do potential biases or low-confidence regions in these predicted structures affect retrieval accuracy and the final sequence design?
3.  For residues lacking a structural match, the model defaults to a uniform prior. Please clarify how high-fidelity predictions are made for these regions without external information.

---

### Note · Authors · 2025-12-05

I have read and agree with the venue's withdrawal policy on behalf of myself and my co-authors.